# Knowledge about Cervical Cancer and Awareness of Human Papillomavirus (HPV) and HPV Vaccine among Female Students from Serbia

**DOI:** 10.3390/medicina56080406

**Published:** 2020-08-13

**Authors:** Nataša K. Rančić, Milan B. Golubović, Mirko V. Ilić, Aleksanda S. Ignjatović, Radomir M. Živadinović, Saša N. Đenić, Stefan D. Momčilović, Biljana N. Kocić, Zoran G. Milošević, Suzana A. Otašević

**Affiliations:** 1Faculty of Medicine Niš, University of Niš, 18 000 Niš, Serbia; drsalea@yahoo.com (A.S.I.); zivadinovicrasa@gmail.com (R.M.Ž.); biljaizzz@yahoo.com (B.N.K.); zormilzoran@gmail.com (Z.G.M.); otasevicsuzana@gmail.com (S.A.O.); 2Clinic of Pediatrics, Clinical Center Niš, 18 000 Niš, Serbia; milanpfc@gmail.com; 3Center for Control and Prevention of Diseases, Institute for Public Health Niš, 18 000 Niš, Serbia; mirkoilic1971@gmail.com; 4Center for Radiology, Clinical Center Niš, 18 000 Niš, Serbia; saska0906@gmail.com; 5Clinic for Reconstructive and Plastic Surgery, Clinical Center Niš, 18 000 Niš, Serbia; m-stefan@mts.rs

**Keywords:** cervical cancer, HPV, HPV vaccine, knowledge, awareness, student population

## Abstract

*Background and objectives*: Persistent infection with carcinogenic human papillomavirus (HPV) is the leading cause of cervical cancer. The study explored students’ knowledge about cervical cancer and awareness of human papillomavirus and the HPV vaccine. *Materials and Methods*: A questionnaire-based survey was carried out among 1616 first-year female college students at the University of Niš. It examined socio-demographic characteristics, measured the score of knowledge about cervical cancer, assessed awareness regarding HPV and the HPV vaccine and inquired about the source of information about cervical cancer and HPV. *Results*: The average cervical cancer knowledge score was 16.35 ± 7.92 (min 0, max 30), with medical professional education, parents’ education level, place of residence and relationship status having significant effects on the score. The awareness about HPV and the HPV vaccine was low, with only 14.2% of students having heard about both HPV and its vaccine. The most commonly reported sources of information were the media, while the most competent one was organized health education. *Conclusions*: Health promotion campaigns and educational programs are necessary in order to reduce cervical cancer burden and should be directed particularly towards those who have demonstrated low cervical cancer knowledge and low awareness regarding HPV and its vaccine.

## 1. Introduction

Malignant disease rates have reached enormous proportions globally and are appropriately considered a modern age plague. Cervical cancer (CC) is the most common gynecological malignancy and has the fourth highest incidence and mortality rate among women worldwide, with about 570,000 cases and 311,000 deaths in 2018 [1]. This dreadful statistic is mostly the result of inefficiently organized health systems in developing and especially in underdeveloped countries. Accordingly, particularly affected by CC are sub-Saharan African and Southeast Asian countries, in which this disease represents the leading cause of cancer death ahead of breast and lung cancer [1].

In Europe, CC causes just over 61,000 new cases per year, which makes it the ninth most common cause of female cancer, and when it comes to cancer deaths, it is ranked 11th. However, it is ranked second in both incidence and mortality among females between 15 and 44 years of age [2]. This trend of reaching peak incidence at younger age, which has been consistantly observed during the past several decades [3], indicates that CC is an age-related disease, especially afflicting women of reproductive age.

This cancer has slow progression [4] and when diagnosed in the early stages it is highly treatable, with a 5-year relative survival rate of up to 92% [5]. Besides, it is the first cancer to have am identified agent essential for its development, human papillomavirus (HPV) infection. Back in the 1970s, zur Hausen suggested a possible association between HPV infection and cervical cancer [6], which was confirmed in the following decades [7]. More than 200 HPV types have been recognized and about 40 of them are sexually transmitted and capable of causing mucosal infection [8,9]. Among them, several types have been recognized as high risk according to their oncogenic potential, with types 16 and 18 being the most prevalent ones [8]. 

Findings from the first study of HPV infection prevalence in Serbian women [9] showed that nearly one third of HPV-positive women in Serbia had types HPV16/18, marked as the most aggressive high-risk genotypes of human papillomavirus HR-HPVs. According to the findings of Kovačević [10], the most prevalent HPV types in females of the Autonomous Province of Vojvodina, in Serbia, are: HPV 16; HPV 31, HPV 51; HPV 33; HPV 18; HPV 52; HPV 56; HPV 39; HPV 45; HPV 58; HPV 59 and HPV 35 and the most frequent HPV types are type 16 and type 31. The most prevalent HPV types of this region showed concordance with European isolates, but non-European variants were also found [10,11].

Viral DNA of these high-risk strains nowadays can be easily detected in exfoliated cervical cells using commercially available tests [12]. This HPV test represents a convenient, highly sensitive screening tool that is rapidly becoming a part of screening guidelines of many countries and has a tendency of becoming the main screening method [13]. Still, it has an important flaw and that is its lower specificity [12]. Apart from this, there is also the well-known Pap smear screening method. It is widely available, cheap and has good specificity for the detection of precancerous lesions with a proven impact on reducing CC prevalence and mortality rates [13]. Organized cervical cancer screening has been conducted in Serbia since 2012 and it covers women aged 25 to 69 years [14]. Alongside these secondary prevention methods, since 2006 there has also been a primary prevention option available—the HPV vaccine. It can be found in bi-, quadri- and nine-valent formulations [10] and the recommendation is that girls should be routinely vaccinated between the ages of 9 and 13 [15]. Although the Advisory Committee on Immunization Practices (ACIP) approved the vaccination of boys aged 9 to 26 in October 2011, it is currently only being implemented in the United States of America (USA), some Canadian provinces, Australia and New Zealand [16].

Recommendations for the use of prophylactic vaccines vary by continent and region and, in most countries, including Serbia, only women are currently vaccinated. It is generally recommended that children and young adults aged 9–26 should be vaccinated [11]. In Serbia, vaccination against HPV infection is not part of the mandatory immunization program [14,15,16,17]. HPV vaccination is recommended for children older than nine years before the first sexual intercourse, and primarily for children in the sixth grade of primary school (12 years of age). Active immunization is carried out with the required number of doses (two or three), depending on the type of vaccine and the age at which it is given [17].

Based on the obtained results on the prevalence of the oncogenic types of HPV in the region of the Autonomous Province of Vojvodina, it can be assumed that the introduction of prophylactic bi-valent and quadri-valent HPV vaccines will reduce by nearly 50% the infection caused by high-risk HPV types 16 and 18. The protective effect of the nine-valent vaccine could be close to 90%. The nine-valent vaccine does not contain HPV 35, 56 or 59 [11], so there is a need to develop an HPV vaccine that will be specific for these HPV types in the Serbian population.

As registered vaccines do not provide protection against all oncogenic HPV types and no vaccine is 100% effective, it is necessary to adhere to national recommendations for cervical cancer screening regardless of vaccine status [18]. 

In addition to chronic HPV infection, other risk factors play an important role in the development of CC as well. Among them, the most notorious are smoking, immunodeficiency, long term use of oral contraceptives, promiscuous behavior and having more than two pregnancies. A family history of CC and other descriptive characteristics, obesity and poor diet are also associated with CC [19,20,21,22].

Although in Serbia there is an organized CC screening program and the HPV vaccine is recommended and available for purchase, CC still remains one of the most prevalent malignancies. According to data from the last available national registry in 2015, it was the fourth most common cancer among women in Central Serbia [23]. In 2018, it was assessed that it causes about 1300 new cases and 550 deaths annually. In addition, Serbia has the highest age-standardized CC mortality rates when compared with other Southeast European countries [2].

The objective of the study was to evaluate knowledge about CC, as well as awareness of HPV and HPV vaccination among first-year female college students from Niš, Serbia. In addition, the sources of students’ information about the subjects were also analyzed. The idea was to identify potential target groups for the implementation of customized health promotion programs.

## 2. Material and Methods

### 2.1. Study Design

The study was designed as a cross-sectional survey conducted between December 2018 and March 2019. Participants were female, first-year college students at the University of Niš during a mandatory medical check-up. All subjects were informed that participation in the study was voluntary and that by completing the questionnaire the consent for publication was given. It was made clear that the data collected would remain anonymous. The approval for the study was obtained from the Ethics Committee of the Faculty of Medicine, Niš (number of decision: 01-10204-11/2012).

A total of 2008 questionnaires were distributed and 1616 fully completed, valid questionnaires were included in the study. 

### 2.2. Questionnaire

Data were collected using a semi-structured questionnaire specially created for this investigation. It was a self-administered questionnaire with three distinct sections (Appendix A).
Appendix A: The first section explored respondents’ socio-demographic characteristics, including age, type of faculty, mother’s and father’s education level, place of residence, type of secondary school completed and financial and relationship status. In addition, participants were asked if they ever had sexual intercourse and, if yes, how many sexual partners they had had.Appendix A: The second section consisted of 22 items divided into three subsections (CC knowledge, HPV awareness, HPV vaccination awareness). CC knowledge was evaluated by a composite score estimated using a total of 18 items regarding risk and protective factors, preventive measures and the outcome of CC. Participants had three possible response options regarding proposed factors (protective factor, risk factor and do not know) and correct answers were coded with either one or two points, deepening factor significance. As for the rest of the questions, the given options were “true”, “false” and “do not know” and correct answers were given two points. The total number of points represented the participant’s CC knowledge score (CC-KS), with higher scores meaning better knowledge. The maximum number of points was 30. All questions were based on data from the relevant literature and information provided by the American Cancer Society [14,15,16]. Awareness about HPV was determined based on whether or not participants had heard about HPV infection. The remaining three items were related to HPV vaccination. Subjects were asked to indicate if they have heard about the HPV vaccine. If the answer was positive, they then answered when the best time to get the HPV vaccine is and if the HPV vaccine is available in Serbia.Appendix A: The last part of the questionnaire inquired about the source of respondents’ knowledge about the subject. Thirteen multiple choice answers, from three categories of source (personal contact, organized health education (OHE) and media) were offered. The opportunity to declare not having any knowledge was also given.

### 2.3. Statistical Analysis

Data are presented as the mean and standard deviation (SD) or as frequencies and proportions. Comparison of the mean values between two groups was done using *t*-tests or Mann–Whitney tests depending on data distribution, while the mean values between the three groups were compared using either ANOVA or Kruskal–Wallis tests. The chi-squared test was used for the comparison of categorical variables. The *p*-value was set at *p* < 0.05 (two-tailed). All statistical analyses were performed using R software, version 3.0.3 [24].

## 3. Results

Overall, 1616 participants completed the whole questionnaire, meaning that the response rate was 80.5%. The average age of the study population was 19.13 ± 0.73 years (min 17, max 29), with the majority of the subjects coming from urban areas (75.7%). Their other socio-demographic characteristics are presented in Table 1. 

### 3.1. Knowledge about CC

As described in the methodology, the knowledge about CC was evaluated using a composite score. It was established that the average score in our students was 16.35 ± 7.92 (min 0, max 30). Statistical analysis showed that medical students had significantly a higher knowledge score compared to other, non-medical students (*p* < 0.001). Pre-college education also significantly influenced participants’ scores (*p* < 0.001), with the highest score noted in students that previously attended medical high school, followed by those that attended grammar schools (Table 2). Furthermore, a significant difference in the score was established between participants from urban and rural areas (*p* = 0.01), with the former being better informed about CC. Mother’s and father’s level of education also influenced the CC knowledge of our subjects (*p* < 0.001 and *p* = 0.006, respectively). The knowledge was significantly higher in students whose mothers had middle (*p* = 0.002) and high (*p* < 0.001) education compared to those with low education. Likewise, having a father with high education meant having higher score (Table 2). The financial status of participants was significantly associated with knowledge score (*p* = 0.008). Thus, participants with high and middle incomes demonstrated higher scores (Table 2). Students in a relationship showed better knowledge than single students (*p* = 0.019). Hohd On the other hand, no significant difference was proven between women that had sexual experience and those who did not (Table 2). It is correct. Those who were in relationship were more aware both of HPV infection and CC.

### 3.2. HPV and HPV Vaccine Awareness 

The total number of participants that had heard of HPV infection was 788 (48.8%), with a significantly higher percentage of medical students being aware of this infection (*p* < 0.001). Furthermore, knowledge of this infection differed significantly depending on the type of secondary education (*p* < 0.001). Thus, students that had finished medical high school had significantly higher knowledge compared to those from grammar school (*p* < 0.001) and non-medical high schools (*p* < 0.001). Finally, students that were in a relationship (*p* = 0.005) and that were sexually active (*p* = 0.001) knew about HPV infection significantly more often (Table 3).

When it comes to HPV vaccine, the overall awareness was quite low (21.3%), whereas an even lower percentage of students had heard of both HPV and its vaccine (14.2%). A statically significant difference was noted relative to mother’s education and relationship and sexual status. Students raised by highly educated mothers had heard of the HPV vaccine more often (compared to students whose mothers had low education, *p* = 0.028), just as the students that were in relationships (vs. single, *p* = 0.041) and that had had sexual intercourse (compared to those who had not had sexual intercourse, *p* = 0.001) (Table 3).

The other two questions about HPV vaccination were correctly answered by only a small number of subjects. Of those who declared that they were aware of HPV and the vaccine, only 37 knew when the best time to get the HPV vaccine is and 48 knew that it is available in Serbia. No difference between distinct demographic groups was established (Table 3).

### 3.3. Source of Information

The media was listed as the dominant source of information about the investigated topics by 1011 of the examined students. The other two categories, organized health education (OHE) and personal contact, were reported as sources by 866 and 538 students, respectively. Two hundred and fifty-five subjects declared having no knowledge about the investigated topics. 

The knowledge score was significantly higher in subjects that received information from any of the three source categories compared to subjects who denied receiving information from those particular categories (Table 4). A similar relationship is noted between information sources and HPV awareness. Among the students who knew about HPV, significantly more reported acquiring information through each of the source categories (vs. not acquiring information through those sources). Finally, OHE was the only information source category associated with a significantly higher number of subjects that had heard about the HPV vaccine (*p* < 0.001), that knew when it should be administered (*p* = 0.039) and that it is available in Serbia (*p* = 0.029).

The complete statistical analysis regarding the sources of information among our participants and their relationship with CC-KS, HPV and the HPV vaccine awareness is shown in Table 4.

## 4. Discussion 

“An ounce of prevention is worth a pound of cure,” an old quote popularized by Benjamin Franklin (1735), makes a lot of sense in medicine [25]. Prevention is especially important in oncology, since cancer is one of the biggest public health issues in the modern world. Paradoxically, despite being highly suitable for thriving preventive programs, CC still represents the most common gynecological cancer worldwide [1]. Therefore, public education is needed for a better understanding of CC, its risk factors and for better disease prevention. Such education programs are only efficient when planned with carefully selected target groups. Aiming to define possible groups for these education programs in Serbia, in this article, we analyzed the knowledge and awareness of first-year female students about CC and HPV. 

Our students demonstrated decent knowledge about CC, with the mean score being more than half of the total possible score. However, in a study done in Serbia’s capital, Belgrade, a considerable portion of women had insufficient knowledge about CC, especially those who did not participate in mass screening [26]. This discrepancy can be explained by differences in the study population, since our research included only college students, the future intellectual elite of the state. Furthermore, a good share of our participants had affiliation with the medical profession, which greatly contributed to a higher CC knowledge level. In that manner, we proved that subjects involved in medical studies or those that attended medical high school had the highest knowledge about CC. Other authors also confirmed that medical students are highly educated on this matter [27,28], which is understandable and very commendable, as they represent the leaders of further education of the general public about CC. In accordance with the previous interpretation, the results of another Serbian study showed that both students of a secondary medical school and midwives who had finished at that same school had significantly better knowledge than patients [29]. Apart from education, we also identified good financial status, living in urban areas and having parents with higher education as factors that contribute to better knowledge about CC. This means that higher socio-economic status is associated with a better health standard and can explain the fact that the highest prevalence of CC is seen in underdeveloped countries [1]. However, data from relevant studies are inconsistent regarding this matter, occasionally showing low knowledge about CC even in people from developed countries. For example, in other Balkan countries, the situation regarding CC and HPV knowledge is generally quite poor. In Romania, even though the majority of interviewed women had heard of HPV, the level of knowledge was low, especially about risk factors for infection [30]. Likewise, Greek adolescents were also insufficiently informed about risk factors and protection methods against CC [31]. A similar situation is also recorded in Hungary and Slovenia [32,33]. The inconsistency of the results from different studies was also noted regarding the association between relationship status and knowledge about CC. While in our study, students in relationship had a higher CC-KS, there are other studies in which being single was associated with knowledge about CC screening [34].

Despite relatively good knowledge about CC in the investigated population, HPV awareness was quite low. We established that a bit less than 50% of the investigated students had heard of HPV infection. Research performed in Slovenia [33] that included women of a wide range of ages and research that included adolescents from Greece [31] also noted low awareness of HPV. On the other hand, studies in the UK [35] and in neighboring Romania [30] showed that between two thirds and three quarters of questioned women had heard about HPV. Such a result was also recorded in our study among subjects that finished medical high school and those attending a medical faculty. Furthermore, these participants had significantly more chance of knowing about HPV, just as other studies established [36]. In addition, in our study, females in relationships and the ones that had sexual experience were more likely to have heard about HPV. This is encouraging since it implies that the subjects who enter into a relationship at least have basic awareness about HPV. Research completed in Germany also found that women that had had sexual intercourse at least once had more often heard of HPV [37].

A significant part of our study was also an investigation of HPV vaccination awareness. This topic was especially important, as CC prevalence in Serbia is very high [22,23]. Unfortunately, the obtained results are disappointing. Although 10 years have passed since the first shot was administered in Serbia, only one fifth of the students knew about the existence of the HPV vaccine and only 14.2% knew about both HPV and its vaccine. Even worse are the findings that, among those who were aware of the vaccine’s existence, only a few knew when it should be administered and that it is available in Serbia. These findings suggest that awareness about the prevention of CC, as an important public health issue, is still at a very low level, which can and should be raised by educating the populace. Another important element in the promotion of CC prevention is introducing the HPV vaccine into the National Immunization Program (NIP). Unfortunately, the HPV vaccine, although recommended, is not yet integrated in the Serbian NIP. On the other hand, this was done years ago in most EU countries, so their good perception about CC prevention is not surprising. In Italy, one of the world’s most developed countries, a remarkably high percentage of adolescents that know about HPV vaccine is observed [38]. Likewise, Serbia’s neighbors, Romania [30] and Hungary [32], as well as nearby Greece [31], also register good awareness of the HPV vaccine, despite the previously mentioned low knowledge. All these countries are in the EU and have included the vaccine in their NIPs, indicating its importance. On the other hand, data from Ukraine, an underdeveloped country that had not introduced the HPV vaccine in its NIP yet, show relatively low awareness of the vaccine among second-year medical students, but still a bit better than among our participants [39]. There were a small number of female adolescents that received the HPV vaccine, after a gynecologist’s recommendation. We do not have the data about vaccinated persons yet and we do not know the exact number of vaccinated persons.

Finally, we investigated the source of participants’ awareness and knowledge. The most frequently chosen source was the media with, 1011 students answering that this source contributed to their knowledge on the subject. Likewise, in another Serbian study, participating females identified the media as the main source of information about CC risk factors [40]. This is a substantial change from previously reported results that the media is a “poor source of comprehensive information on cervical cancer” [41]. Two separate pieces of research from Hungary and Slovenia also found that the media, namely the internet and magazines, to be dominant sources of information about this topic [32,33,42]. This confirms that the media has a great impact on public opinion and thus can be a powerful tool for increasing the awareness of CC. This source was also the most commonly selected among students that had heard about HPV, just as it was in an American study that also included young women [43]. Like so, those who selected the media as source of information had a significantly greater chance of knowing about HPV. In addition, a statistically significant difference in the CC-KS was established between participants who had been informed through the media and those who had not. Unfortunately, the knowledge score was more than one point lower than the score of participants who chose OHE or personal contact as a source. Such a result indicates that, although information from the media is important, it often does not have enough credibility and can even, if not critically assessed, produce counter effects and mislead the audience. Therefore, the accuracy of the information is crucial. Furthermore, the finding that there is no significant difference in HPV vaccination awareness between those who did and did not select the media as a source indicates that future media campaigns should be especially focused on this topic. Messages about HPV infection and HPV vaccination and its importance for the prevention of CC conveyed through media—to parents, providers, policymakers and the general public—may contribute to a rise in vaccination coverage. Regarding the competence of the source, according to the results of this study, OHE is the most efficient one, with the CC-KS being the highest in the participants who picked it. Besides, OHE significantly affected the awareness of HPV and was the only source that was associated with a significant likelihood of participants having knowledge about the HPV vaccine. While this is an expected result, it is more than welcome since it shows that OHE represents a trustworthy source, which should have a dominant role in educational programs. Other surveys also showed that physicians are the most desirable source of information, which is highly positive and means patients also recognize OHE as a qualified and authoritative source [44,45]. The third possible way of gaining information about CC was interpersonal communication with friends or family, which in our survey produced paradoxical results. Students that acquired knowledge through this source had a significantly higher CC-KS. On the other hand, a significantly smaller number of subjects that knew about HPV marked friends or family as an information source, while having knowledge about the vaccine was not associated with this source. Accordingly, although obtaining information from family or friends is easy, convenient and often the most common source of the subjects’ knowledge [45] our data suggest that it should not have the main role in education. 

## 5. Conclusions

The present study revealed sufficient knowledge about CC in the examined population, but awareness about HPV infection was rather low. The situation about HPV vaccine awareness was even more disappointing, since only 14.2% had heard of both HPV and the HPV vaccine. Moreover, only a negligible number of students had knowledge about the best time to get the HPV vaccine and that the vaccine is available in Serbia. Since the highest score of knowledge about CC was determined in the group of students who denoted organized health education as a source of information, this established scenario should serve as a guide for the development of high-quality education programs directed particularly towards raising awareness about CC prevention and the importance of HPV vaccination. It can be recommended that a special focus should be on students whose commitments in education are not related to health sciences. Furthermore, in accordance with the result that the media was the most common source of information, these campaigns should be conducted using exactly this information source. However, steps must be taken in order to increase its reliability, thus making the media a powerful instrument through which correct information could be spread to general public. In addition to such well-targeted health promotion campaigns, an official national strategy is also necessary with the goal to introduce the HPV vaccine in the NIP, with monitoring and real support from authorities. Only by achieving these goals can the CC burden be reduced.

## Figures and Tables

**Table 1 medicina-56-00406-t001:** Socio-demographic characteristics of the participants.

Variable	N	(%)
**Faculty/college attending**		
Medical	229	(14.2)
Other	1387	(85.2)
**Type of high school finished**		
Grammar school	665	(41.1)
Other, non-medical high school	709	(43.9)
Medical high school	242	(15)
**Place of residence**		
Urban area	1224	(75.7)
Rural area	392	(24.3)
**Mother’s level of education ^a^**		
Low	81	(5)
Middle	1000	(62)
High	531	(33)
**Father’s level of education ^a^**		
Low	47	(2.9)
Middle	940	(59.1)
High	604	(38)
**Financial status/income**		
Low	159	(9.8)
Middle	825	(51.1)
High	632	(39.1)
**Marital/relationship status ^b^**		
Single	890	(55.1)
In relationship	726	(44.9)
**Ever had sexual intercourse**		
Yes	820	(50.7)
No	796	(49.3)
**Number of sexual partners ^c^**		
1	526	(64.1)
2–4	272	(33.2)
5+	22	(2.7)

^a^ Missing data due to death of a parent; ^b^ Other response options were not chosen by our participants; ^c^ Of those that had had sexual intercourse; 5+—five and more.

**Table 2 medicina-56-00406-t002:** Association of participants’ socio-demographic characteristics with cervical cancer knowledge score.

Variable	Cervical Cancer	*p*
	Knowledge Score	
**Faculty/college attending**		<0.001
Medical	21.21 ± 5.73	
Other	15.54 ± 7.94	
**Type of high school finished**		<0.001
Grammar school	16.79 ± 7.73	
Other, non-medical high school	14.69 ± 7.98 ^a,b^	
Medical high school	20.82 ± 6.38 ^b^	
**Place of residence**		0.01
Urban area	16.62 ± 7.82	
Rural area	15.48 ± 8.16	
**Mother’s level of education**		<0.001
Low	13.33 ± 8.27 ^c,d^	
Middle	16.22 ± 8.04	
High	17.02 ± 7.53	
**Father’s level of education**		0.006
Low	15.68 ± 7.82	
Middle	15.78 ± 8.09	
High	17.22 ± 7.57 ^e^	
**Financial status/income**		0.008
Low	14.25 ± 8.53	
Middle	16.62 ± 7.85 ^f^	
High	16.51 ± 7.79 ^f^	
**Marital/relationship status**		0.019
Single	15.89 ± 8.06	
In relationship	16.96 ± 7.69	
**Ever had sexual intercourse**		0.065
Yes	16.67 ± 7.90	
No	16.01 ± 7.93	
**Number of sexual partners ^g^**		0.537
1	16.65 ± 7.80	
2–4	16.96 ± 8.08	
5+	17.00 ± 8.45	

^a^ Vs. medical high school, *p* < 0.05; ^b^ vs. grammar school, *p* < 0.05; ^c^ vs. middle educated mother, *p* < 0.05; ^d^ vs. high educated mother, *p* < 0.05; ^e^ vs. middle educated father, *p* < 0.05, ^f^ vs. low income, *p* < 0.05; ^g^ of those that had had sexual intercourse; 5+—five and more.

**Table 3 medicina-56-00406-t003:** Association of participants’ socio-demographic characteristics with HPV and HPV vaccine awareness.

	Heard about HPV *N* (%)		Heard about HPV Vaccine ^a^ *N* (%)		Know When the Best Time to Get the HPV Vaccine is ^b^ *N* (%)		Know that HPV Vaccine is Available in Serbia ^b^ *N* (%)	
**Faculty/college attending**								
Medical	155	67.7	50	32.3	10	6.5	15	9.7
Other	633	45.6	179	28.3	27	4.3	33	5.2
*p* *	<0.001		0.379		0.249		0.037	
**Type of high school finished**								
Grammar school	295 **	44.4	84	28.5	14	4.7	16	5.4
Other, non-medical high schools	303 **	42.7	79	26.1	15	5.0	19	6.3
Medical high school	190	78.5	66	34.7	8	4.2	13	6.8
*p* *	<0.001		0.119		0.930		0.805	
**Place of residence**								
Urban area	603	49.3	172	28.5	29	4.8	38	6.3
Rural area	185	47.2	57	30.8	8	4.3	10	5.4
*p*^1,^*	0.512		0.612		0.785		0.656	
**Mother’s level of education ^c^**								
Low	43	53.1	6 ***	14.0	1	2.3	0	0.0
Middle	475	47.5	138	28.8	20	4.2	26	5.5
High	268	50.5	85	31.7	16	6.0	22	8.2
*p* *	0.394		0.041		0.415		0.074	
**Father’s level of education ^c^**								
Low	20	42.6	8	40.0	1	5.0	1	5.0
Middle	466	49.6	133	28.5	18	3.9	26	5.6
High	295	48.8	86	29.2	18	6.1	20	6.8
*p* *	0.634		0.561		0.366		0.780	
**Financial status/income**								
Low	73	45.9	22	30.1	4	5.5	3	4.1
Middle	388	47	110	28.4	18	4.6	31	8.0
High	327	57.1	97	29.7	15	4.6	14	4.3
*p* *	0.153		0.908		0.946		0.090	
**Marital/relationship status**								
Single	406	45.6	105	25.9	14	3.4	20	4.9
In relationship	382	52.6	124	32.5	23	6.0	28	7.3
*p* *	0.005		0.041		0.088		0.159	
**Ever had sexual intercourse**								
Yes	435	53.0	148	34.0	21	4.8	31	7.1
No	353	44.3	81	22.9	16	4.5	17	4.8
*p*^1,^*	0.001		0.001		0.846		0.177	
**Number of sexual partners ^d^**								
1	274	52.1	97	35.4	15	5.5	19	6.9
2–4	149	54.8	45	30.2	4	2.7	11	7.4
5+	12	54.5	6	50.0	2	16.7	1	8.3
*p* *	0.763		0.285		0.067		0.972	

* Chi-squared test; ** vs. medical high school, *p* < 0.05; *** vs. high educated mother, *p* < 0.05; ^a^ only those that had also heard about HPV were taken into account; ^b^ only those that had heard about HPV and the HPV vaccine were taken into account; ^c^ missing data due to death of a parent; ^d^ of those that had had sexual intercourse; ^1^—not significant; 5+—five and more; Human papillomavirus—HPV.

**Table 4 medicina-56-00406-t004:** Source of information and its association with cervical cancer knowledge score, HPV and HPV vaccine awareness.

Source of Information	Knowledge Score about Cervical Cancer	Heard about HPV		Heard about HPV Vaccine ^a^		Know the Best Time to Get the HPV Vaccine ^b^		Know that HPV Vaccine is Available in Serbia ^b^	
	*N*	*N*	*(%)*	*N*	*(%)*	*N*	*(%)*	*N*	*(%)*
**Media**									
Yes	17.38 ± 7.52	532	67.5	158	69.0	29	78.4	34	70.8
No	14.62 ± 8.28	256	32.5	71	31.0	8	21.6	14	29.2
*p*	<0.001 *	<0.001 ***		0.628 ***		0.249 ***		0.893 ***	
**OHE** ^c^									
Yes	18.49 ± 7.15	512	65.0	170	74.2	33	89.2	42	87.5
No	13.87 ± 8.05	276	35.0	59	25.8	4	10.8	6	12.5
*p*	<0.001 *	<0.001 ***		<0.001 ***		0.039 ***		0.029 ***	
**Personal contact**									
Yes	18.42 ± 6.98	298	37.8	94	41.0	17	45.9	20	41.7
No	15.31 ± 8.16	490	62.2	135	59.0	20	54.1	28	58.3
*p*	<0.001 **	<0.001 ***		0.264 ***		0.632 ***		0.922 ***	

* Mann–Whitney test; ** independent *t-*test; *** chi-squared test; ^a^ Only those who had also heard about HPV were taken into account; ^b^ Only those who had heard about HPV and the HPV vaccine were taken into account; ^c^ OHE—organized health education.

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
