# Peer review of "Knowledge about Cervical Cancer and Awareness of Human Papillomavirus (HPV) and HPV Vaccine among Female Students from Serbia"

_medicina, 2020, doi:10.3390/medicina56080406_

Round 1

Reviewer 1 Report

In this manuscript by Rancic et al., the authors examined the inter-relationship about the knowledge of cervical cancer and awareness of HPV as an etiological agent of cervical cancer and the availability of the HPV vaccine among young females in Serbia. This is an important study, as enough research and literature is currently available for educating women worldwide, that could significantly reduce the numbers of  deaths from this devastating cancer.  The study asked some critical questions  regarding women’s perspective on HPV and cervical cancer. The fact that the authors took into account education levels of family members is of significance. The report is well written. I would like to see the following information added to the writing, where it is possible to add to the flow of discussion.

[1] What prevalent HPV types are present in the Serbian population ?

[2] Are the bi- and quadrivalent HPV vaccines currently available effective in targeting the prevalent HPV types present in the Serbian population ?

[3] Is there a need to develop HPV vaccines that are specific for targeting the HPV types in this population ?

[4] What is known about HPV vaccination of men in this population ? What are the current views of men receiving vaccination in this population ?

[5] Does Serbia have any government policies for implementing HPV vaccinations in this population ? If so, who enforces these policies?

[6] Is there health care coverage for women to receive HPV vaccinations at the appropriate times and is there screening available for early detection ?

Author Response

Findings from the first study of HPV infection prevalence in Serbian women showed that mearly one third of HPV- positive women in Serbia had HPV 16/18 types marked as the most agressive HR-HPVs. Acvording to the findings of Kovačević the most prevalent HPV types in females of the Autonomous Province of Vojvodina, in Serbia are: HPV 16, HPV 31, HPV 51, HPV 33, HPV 18, HPV 52, HPV 56, HPV39, HPV 45,HPV58,HPV59,and HPV 35. The most frequent HPV types 16 and type 31.  Although in October 2011, the Advisory Commitee on Immunization Practice approved the vaccination of boys aged of 9 to 26, it is currently being implemented in the Unated States of America, some Canadian provinces, Australia and New Zealand. In the most countries, includig Serbia, only women are currently vaccinated. In Serbia, HPV vaccination is not part of the mandatory Immunization program and it id recommended for children olfe than nine years before the first sexual intercourse. and primarly for children in the sixth grade of primary school, at 12 years of age. It canbe assumed that the introduction of bi-valent a d quadri-valent vaccines  will reduce nearly 50% infection caused by high-risk HPV types 16 and 18. The protective effect of the nine-valent vaccine could be close to 90 %. The nine-valent vaccine does not containts HPV types 35, 56 and 59 so there is a need to develope HPV vaccine that will be specific for these HP V types. As registered vaccines do not provide protection against all oncogenic HPV types a d no vaccine is 100% effective it is necessary to adhere to cevical cancer screening regardless of vaccine status. The HPV testing is highly sensitive screening tool and in Serbia i, a well -known Pap smear screening method is mandatory hor women aged 25-69 years of  age.

,

Reviewer 2 Report

It would be very interesting if the authors could comment about the number of vaccinated women in their country with HPV.

Could the authors comment on how did they establish the rating points for each answer as it is shown in the example of the questionnaire

Did the authors do any research regarding the raising knowledge of the respondents about HPV vaccine after completing the questionnaire.

Following the conclusion that media is most used for information how do the author plan to use this information for raising awareness

Author Response

Scoring. Cervical cancer-CC,  knowledge was evaluated by composite scores estimated using a total items regardig risk and protective factors, protective measures and outcome of CC. Participants had three possible response options regarding priposed factors (protective factor, risk factor, and don't know) and correct answers were coded with either one or two points depending on factor significance. As for the rest of the questions, the given options were "true", "false" and "don't know" and correct answers were given two points. The total number of points represented the participant's CC knowledge score (CC-KS) with higher scores meaning better knowledge. The maximum number of points was 30. All questions were based on data from relevant literature and information provided by American Cancer Society.

There are small number of vaccinated females, adolescent in the city of Niš. I don't  have this data.

Messages about HPV infection and HPV vaccination and it's imortance for prevention of cervical cancer througt media- to parents, providers, policymakers, and the general public may contribute to rise vaccination coverage.

Round 2

Reviewer 1 Report

Thank you for the revised manuscript.

This manuscript is a resubmission of an earlier submission. The following is a list of the peer review reports and author responses from that submission.